# Antibiotic Resistance Genes Associated with Marine Surface Sediments: A Baseline from the Shores of Kuwait

**Nazima Habibi** [1,*] , **Saif Uddin** [1,*] , **Brett Lyons** [2] , **Hanan A. Al-Sarawi** [3] , **Montaha Behbehani** [1] , **Anisha Shajan** [1] , **Nasreem Abdul Razzack** [1] , **Farhana Zakir** [1] **and Faiz Alam** [1]

1. Environment and Life Science Research Centre, Kuwait Institute for Scientific Research, Safat 13109, Kuwait; mbahbaha@kisr.edu.kw (M.B.); ashajan@kisr.edu.kw (A.S.); nabdulr@kisr.edu.kw (N.A.R.); fhussain@kisr.edu.kw (F.Z.); fshirshikhar@kisr.edu.kw (F.A.)
2. Centre for Environment, Fisheries and Aquaculture Science, The Nothe, Barrack Road, Weymouth, Dorset DT4 8UB, UK; brettlyons1@hotmail.com
3. Research and Studies Office, Environment Public Authority, Shuwaikh Industrial Area, Safat 13104, Kuwait; h.alsarawi@gmail.com
* Correspondence: nhabibi@kisr.edu.kw (N.H.); sdin@kisr.edu.kw (S.U.)

**Abstract:** Marine sediments are a sink for antibiotic resistance genes (ARGs) and antibiotic-resistant microbes (ARMs). Wastewater discharge into the aquatic environment is the dominant pathway for pharmaceuticals reaching aquatic organisms. Hence, the characterization of ARGs is a priority research area. This baseline study reports the presence of ARGs in 12 coastal sediment samples covering the urban coastline of Kuwait through whole-genome metagenomic sequencing. The presence of 402 antibiotic resistance genes (ARGs) were recorded in these samples; the most prevalent were patA, adeF, ErmE, ErmF, TaeA, tetX, mphD, bcrC, srmB, mtrD, baeS, Erm30, vanTE, VIM-7, AcrF, ANT4-1a, tet33, adeB, efmA, and rpsL, which showed resistance against 34 drug classes. Maximum resistance was detected against the beta-lactams (cephalosporins and penam), and 46% of genes originated from the phylum Proteobacteria. Low abundances of ESKAPEE pathogens (*Enterococcus faecium*, *Staphylococcus aureus*, *Klebsiella pneumonia*, *Acinetobacter baumanii*, *Pseudomonas aeruginosa*, *Enterobacter* sps., and *Escherichia coli*) were also recorded. Approximately 42% of ARGs exhibited multiple drug resistance. All the ARGs exhibited spatial variations. The major mode of action was antibiotic efflux, followed by antibiotic inactivation, antibiotic target alteration, antibiotic target protection, and antibiotic target replacement. Our findings supported the occurrence of ARGs in coastal marine sediments and the possibility of their dissemination to surrounding ecosystems.

**Keywords:** shotgun metagenomics; marine sediments; antibiotic resistance genes; antimicrobial resistance microbes

## 1. Introduction

About 40% of the world's population lives within 100 km of the coast. The highly developed coastline of Kuwait is known to have drained land-based pollutants into the coastal environment [1–16]. Metals, organic contaminants, naturally occurring radioactive material (NORMs), pharmaceutical compounds, microplastics, and others are discharged, leaked, and leached from ports, shipping, oil refineries, desalination plants, wastewater treatment plants, and other industries into the aquatic environment. In addition, contaminants are also transported over a long range with dust, which is a chronic problem in arid regions [17–19]. The coastal environment is one of the most productive ecosystems and harbors a rich microbial community. Recent investigations have suggested that there are $1.7 \times 10^{28}$–$5.4 \times 10^{29}$ microbial cells present in the top 10–50 cm of the sediment profile [20,21]. The presence of pharmaceutical compounds, along with antibiotics in Kuwait's coastal waters [22] and wastewater effluent [23], has raised concerns regarding antibiotic resistance in microbes. Some recent reports have reaffirmed that sediments are reservoirs of

antibiotic resistance genes (ARGs) that can be disseminated in marine environments [24–27]. The aquatic sediments are known to represent an important environmental matrix within which genetic transfer of antimicrobial resistance (AMR) can take place [27,28]. Antibiotic resistance in microbes is reported in estuaries, sediments in coastal areas, and deep marine sediments [29–32].

Twelve bacterial families have been listed by the World Health Organization (WHO) as a threat to human health [33]. Quantitative microbial risk assessment (QMRA) is recommended as an efficient tool for evaluating and quantifying human health risks associated with ARGs [24]. This study attempted to establish the baseline of the ARGs in the microbial community within Kuwait's marine sediments. The DNA samples extracted from the marine sediments were subjected to shotgun metagenomic profiling, taxonomic distribution, and calculation of the abundance of antibiotic resistance genes (ARGs).

## 2. Materials and Methods

### 2.1. Sample Collection and DNA Extraction

A total of 12 surface sediment samples were collected from the Kuwait Marine area (Figure 1) during September–October 2021 (Table 1). These sampling locations were in proximity to stormwater outfalls, which often fugitively discharge wastewater. A grab sample was collected in sterile 50 mL centrifuge tubes covering a 10–15 cm deep sediment profile. Two sites (S4 and S12) that were relatively pristine and free from waste discharges were also sampled. The samples were packed and transported on ice to KISR laboratories. Sample aliquots were stored at −20 °C until DNA extraction. The total DNA from each 0.25 g sample was extracted using a PowerSoil DNA Extraction Kit (QIAGEN, Germantown, MD, USA) [34]. Multiple aliquots from the same site were used for DNA extraction and pooled to reach the desired concentration. The quantity and quality of isolated DNA were evaluated using a Qubit fluorometer (Thermo Fisher Scientific, Waltham, MA, USA) and agarose gel electrophoresis (Bio-Rad, Darmstadt, Germany), respectively. The DNA recoveries from the pristine sites were lower compared to those from other sites. We further estimated the bacterial cell counts at each location through quantitative polymerase chain reaction (qPCR). Universal 16S rRNA primers were employed for this purpose [35]. The PCR reaction was assembled in a volume of 20 μL as per the method described in Habibi et al. [36]. The Ct values were used for the estimation of cells per gram of sediment samples [37]. The DNA concentrations and cell counts are presented in Table 1. Relative to the lowest DNA yield at S4, the cell counts were also minimum. The highest cell counts were obtained at S8, followed by S9 and S10.

### 2.2. Metagenomic Sequencing

A total of 12 metagenomes were sequenced. The NEBNext® UltraTM DNA library preparation kit (Illumina, San Diego, CA, USA) was used to construct DNA libraries. Briefly, 1000 ng of qualified DNA was sonicated to produce 350 bp fragments. These short DNA segments were then end-repaired, A-tailed, and subjected to index PCR [38]. Amplified libraries were purified through Agencourt AMPure XP magnetic beads (Beckman Coulter Genomics, Brea, CA, USA) and quantified through qPCR [39]. The average library size was determined using an Agilent 2100 Bioanalyzer (Agilent Technologies, Santa Clara, CA, USA) [40]. Sequencing was performed on the Illumina NovaSeq 6000 platform using $2 \times 150$ bp paired-end read chemistry. The base percentage composition and quality distribution of the base for each library is provided in Supplementary S1. The raw reads with low-quality bases (Q-value $\leq 38$ and N nucleotides) were trimmed and aligned using Bowtie2 v2.2.4 [41]. Clean reads were assembled using MEGAHIT v1.0.4 into scaftigs. Scaftigs ($\geq 500$ bp) were used for open reading frame (ORF) prediction using MetaGene-Mark v2.10 [42]. The CD-HIT software v4.5.8, (Weizhongli Lab, J Craig Venter Institute, La Jolla, CA, USA) was used to obtain the gene catalog from the filtered ORFs (>100 nt) [43]. Gene catalogs (Unigenes) of the predicted ORFs were obtained by mapping using Bowtie2 v2.2.4 [44].

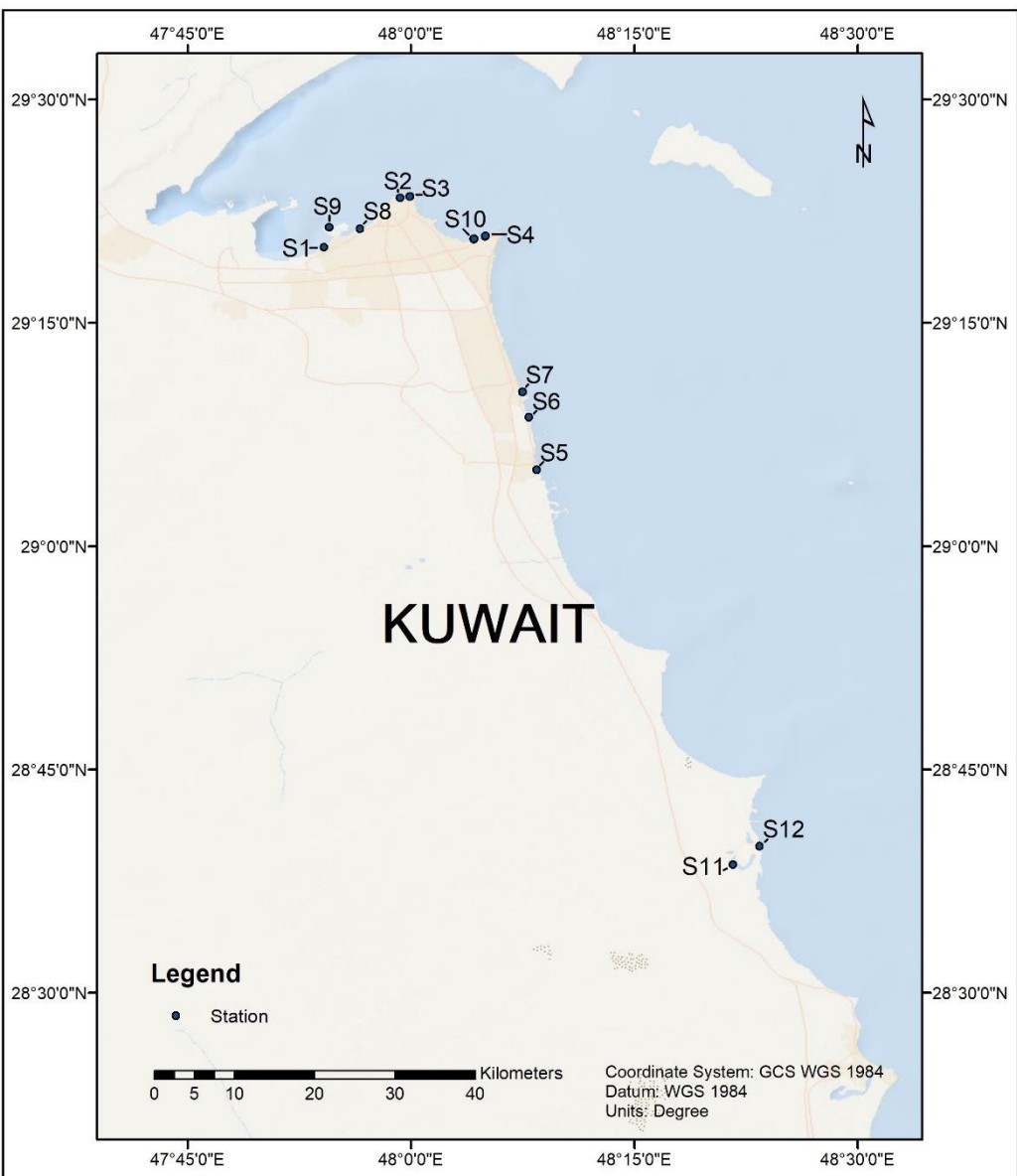

**Figure 1.** Location map showing the sampling locations along Kuwait's coastline.

*2.3. Analysis of Antibiotic Resistance Genes*

The Resistance Gene Identifier (RGI) software was used to align the Unigenes to the Comprehensive Antibiotic Research Database (CARD). The BLASTP values were set as per the standard parameters (e value $\leq 1 \times 10^{-5}$) to filter antibiotic resistance genes (ARGs) [45]. Metastats were used for differential abundance testing at different taxonomic levels by applying the Benjamini and Hochberg false-discovery rate (FDR-q < 0.05) [46]. DIAMOND (v0.9.9) was used to align the Unigenes to MicroNR (v2018-01-02; blastp; e value $\leq 1 \times 10^{-5}$) to filter bacterial taxa associated with the ARGs [47]. The Circos plot was drawn using RCircos [48]. The hierarchical clustering was performed on Euclidean distances using the Ward algorithm [17]. The double-doughnut charts were created in Microsoft Excel® for Mac v16.62, and the Venn diagrams were created in Lucid charts.

**Table 1.** Details of sampling sites, DNA recoveries, and bacterial cell counts.

| Site/Sample Code | Site Description | GPS Coordinates | Date of Sample Collection | DNA Yield (μg g$^{-1}$) Mean ± SD | Cells g$^{-1}$ Sediment Mean |
|---|---|---|---|---|---|
| S1 | KISR Outfall | 29.334824 N, 47.902379 E | 16.09.2021 | 3.63 ± 0.80 | $4.07 \times 10^4$ |
| S2 | Sharq Fisherman's village | 29.390016 N, 47.987360 E | 16.09.2021 | 2.22 ± 0.98 | $6.28 \times 10^4$ |
| S3 | Kuwait towers | 29.391912 N, 47.998332 E | 16.09.2021 | 0.72 ± 0.25 | $3.49 \times 10^7$ |
| S4 | Marina Beach | 29.346564 N, 48.080955 E | 16.09.2021 | 0.03 ± 0.00 | $1.34 \times 10^3$ |
| S5 | Fahaheel | 29.085652 N, 48.140467 E | 16.09.2021 | 0.53 ± 0.24 | $3.75 \times 10^6$ |
| S6 | Mahboula | 29.144615 N, 48.131618 E | 16.09.2021 | 0.67 ± 0.28 | $6.03 \times 10^6$ |
| S7 | Fintas/Eaigila | 29.172970 N, 48.124410 E | 16.09.2021 | 0.96 ± 0.38 | $1.08 \times 10^7$ |
| S8 | KPC beach | 29.355843 N, 47.942600 E | 01.10.2021 | 2.02 ± 0.38 | $8.03 \times 10^9$ |
| S9 | Kuwait free-trade zone | 29.356870 N, 47.908080 E | 01.10.2021 | 3.05 ± 0.66 | $7.64 \times 10^8$ |
| S10 | Marina Main Outfall | 29.347422 N, 48.083147 E | 01.10.2021 | 1.96 ± 0.46 | $4.88 \times 10^8$ |
| S11 | Khairan Fisherman's village | 28.643537 N, 48.360438 E | 13.10.2021 | 2.24 ± 0.57 | $3.52 \times 10^6$ |
| S12 | Khairan inlet | 28.664389 N, 48.389889 E | 13.10.2021 | 0.11 ± 0.06 | $3.12 \times 10^6$ |

Means are average of five subsamples used for DNA isolation from each site.

## 3. Results

Twelve libraries were constructed to study the metagenomes of environmental DNA isolated from Kuwait's marine sediments. The sequences were annotated against the CARD database, and the relative abundance of ARGs was analyzed. Common bacterial phyla, ESKAPEE (*Enterococcus faecium*, *Staphylococcus aureus*, *Klebsiella pneumonia*, *Acinetobacter baumanii*, *Pseudomonas aeruginosa*, *Enterobacter* sps., and *Escherichia coli*) pathogens, and ARG drug classes and their mode of action were classified. Spatial variations in the predominant ARGs were also studied.

### 3.1. Metagenomic Sequencing and Assembly

The sequenced reads were between 5938 and 6875 (Phred > Q20 for 97% of bases), with an average of 6441 reads per sample. Quality filtering and trimming retained 5933 to 6846 reads, with an average of 6432 reads processed per sample. Each sample was de novo assembled into scaftigs ranging from 63,498,148 to 283,332,790 bp (Table 2). The N50 of the assembled genomes ranged from 681 to 1096, and the lengths of the smallest and largest scafftig were 11,228 and 564,351 bp, respectively.

**Table 2.** Metagenome assembly statistics of the marine sediment samples of Kuwait.

| Sample ID | Total Length (bp) | Number | Average Length (bp) | N50 Length (bp) | N90 Length (bp) | Max Length (bp) |
|---|---|---|---|---|---|---|
| S1 | 273,079,570 | 292,818 | 932.59 | 930 | 553 | 65,370 |
| S2 | 283,332,790 | 285,490 | 992.44 | 997 | 556 | 90,537 |
| S3 | 144,882,306 | 175,136 | 827.26 | 782 | 535 | 64,264 |
| S4 | 170,434,411 | 206,843 | 823.98 | 776 | 533 | 118,582 |
| S5 | 188,487,038 | 215,400 | 875.06 | 835 | 538 | 49,103 |
| S6 | 248,200,658 | 234,528 | 1058.30 | 1096 | 561 | 564,351 |
| S7 | 126,649,852 | 162,005 | 781.77 | 739 | 529 | 33,109 |
| S8 | 162,644,191 | 194,236 | 837.35 | 797 | 536 | 30,935 |
| S9 | 137,440,619 | 166,332 | 826.3 | 784 | 533 | 28,815 |
| S10 | 128,300,180 | 159,858 | 802.59 | 755 | 530 | 29,166 |
| S11 | 63,498,148 | 87,889 | 722.48 | 681 | 523 | 32,735 |
| S12 | 75,920,164 | 105,385 | 720.41 | 683 | 523 | 11,228 |

Total length—length of all scafftigs; Number—total number of scafftigs; average length—average length of all the scafftigs; N50—shortest sequence length at 50% of the genome; N90—shortest sequence length at 90% of the genome; Max length—maximum length of the scafftigs.

### 3.2. Antibiotic Resistance Gene Profiles

In total, 402 ARGs (Table S1) were detected in the sediment samples from Kuwait, of which the top 20 are shown in Figure 2. The mean relative abundance (RA%) was highest for patA (3.05 ± 3.6), followed by adeF (mean 2.59 ± 5.5), rpsL (mean 1.80 ± 1.4), TaeA (mean 1.39 ± 2.3), AcrF (mean 1.35 ± 1.7), ErmF (mean 1.14 ± 3.2), mphD (mean 0.98 ± 2.1), vanTE (mean 0.92 ± 1.7), adeB (mean 0.81 ± 1.2), bcrC (mean 0.80 ± 1.4), baeS (mean 0.77 ± 1.6), tetX (mean 0.72 ± 2.2), ErmE (mean 0.71 ± 2.5), Erm30 (mean 0.68 ± 1.7), tet33 (mean 0.66 ± 1.1), srmB (mean 0.65 ± 1.8), mtrD (mean 0.65 ± 1.7), VIM-7 (0.51 ± 1.3), ANT4-la (mean 0.40 ± 1.4), and efmA (mean 0.39 ± 1.3). These genes belonged to 13 AMR gene families, such as ATP-binding cassette, resistance–nodulation–cell division, Erm23S ribosomal RNA methyltransferase, tetracycline inactivation enzyme, macrolide phosphotransferase, undecaprenyl pyrophosphate related proteins, ABC-F ATP-binding cassette, glycopeptide resistance gene cluster-vanT, VIM beta-lactamase, ANT (4′), and the major facilitator superfamily.

The RA and dominance of these genes varied at each site (Figure 3). The genes mtrD, patA, ErmF, and mphD were predominant at S1. At S2, adeF and TaeA were more common. The sampling site of S3 harbored more of patA, TaeA, bcrC, ErmF, and adeF. The abundant ARGs at S4 were patA, srmB, bcrC, and adeF. Genes such as tetX, TaeA, patA, bcrC, and adeF were prevalent at S5. The genes mphD, srmB, mtrD, adeF, tetX, adeF, patA, and TaeA were commonly found at S6. The site S7 exhibited the domination of adeF, TaeA, patA, and tetX. S8 showed a prevalence of patA, bcrC, and TaeA. S9 was richer in ErmF, TaeA, patA, and adeF. Site S10 was a reservoir of patA, mphD, adeF, and bcrC. The site S11 more commonly housed TaeA, adeF, patA, and ErmE genes. The relatively pristine site of S12 also possessed ARGs such as patA, bcrC, and mtrD.

### 3.3. Major Contributing Phyla

The major phyla identified in the bottom sediments were Proteobacteria, Bacteroidetes, Actinobacteria, Cyanobacteria, Firmicutes, Acidobacteria, Balneolaeota, Thaumarchaeota, and others, contributing 53%, 12%, 4%, 4%, 1%, 1%, 1%, 1%, and 23% respectively. Figure 4 presents the relative abundance of ARGs in each phylum.

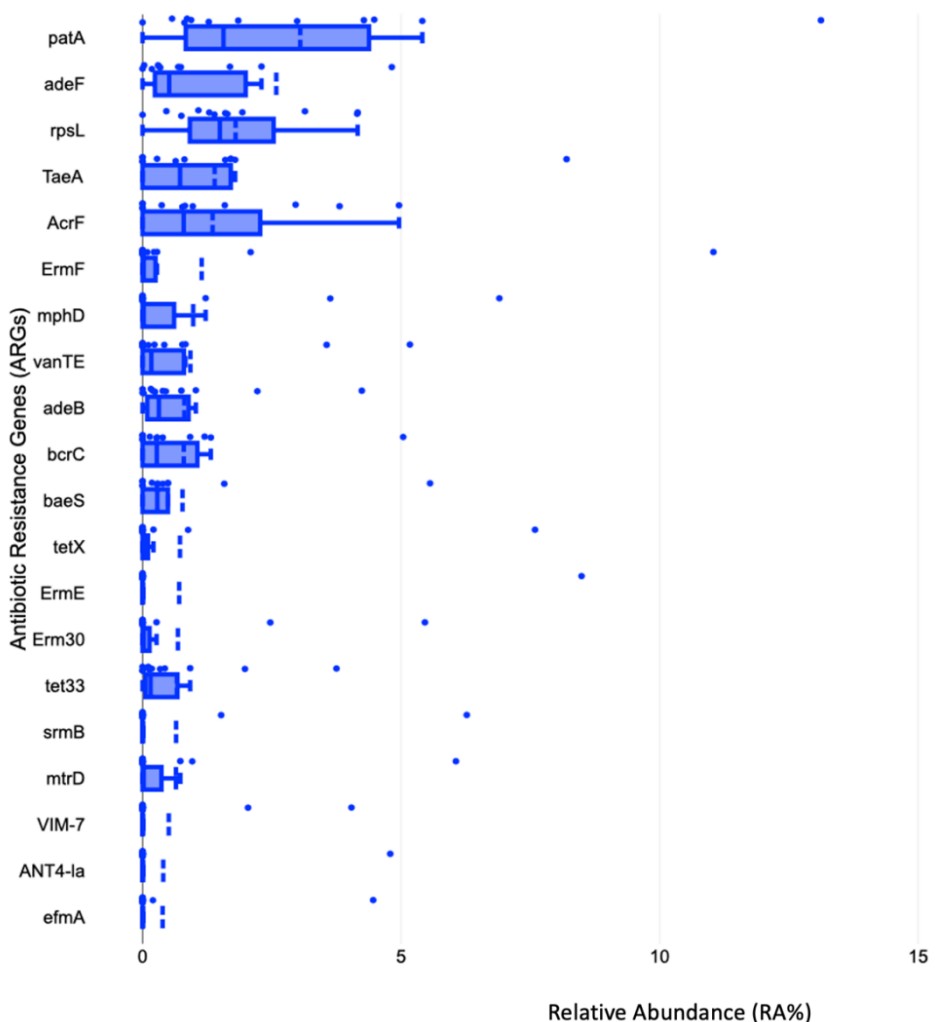

**Figure 2.** Relative abundance (RA%) of predominant ARGs in the marine sediments of Kuwait's coast. For each ARG, a box-and-whisker plot is shown. Each box represents the interquartile range (25–75%), upper and lower whiskers are −10–90%, and dashed blue lines mean RA%. The blue dots denote the RA at 12 sampling points. The x-axis shows the relative abundance, and the corresponding ARGs are plotted on the *y*-axis.

The *Enterococcus faecium*, *Staphylococcus aureus*, *Klebsiella pneumonia*, *Acinetobacter baumanii, Pseudomonas aeruginosa, Enterobacter* sps., and *Escherichia coli* (ESKAPEE) pathogens, known for their multidrug-resistant nature, were also detected in the collected samples (Figure 5). Their relative abundances were very low (>0.01%). The highest average abundances were recorded for *E. coli*, followed by *P. aeruginosa* > *A. baumanii* > *E. faecium*, > *K. pneumonia*, *S. aureus,* and *Enterobacter* sps.

The ESKAPEE pathogens also comprised the coliforms (*Enterobacter* sps., *K. pneumonia*, and *E. coli*) and the enterococci (*E. faecium*). The absolute abundance of these genera in terms of the OTU counts was examined in the sediments from each location. The OTUs of fecal coliform (*E. coli)* were detected all across Kuwait's coast (mean = 696). Comparatively higher counts (>1000 OTUs) were recorded at S1 and S9. The lowest counts were documented from S2 and S12 (~60 OTUs). Among the nonfecal coliforms, the OTUs of *Enterobacter* sps. (mean = 96) and *K. pneumonia* (mean = 94) were also found at all the locations. The former had the maximum at S12 (*n* = 841) and the minimum (*n* = 0) at S2, S6 and S9. *K. pneumonia* had the maximum at S12 (*n* = 362) and the minimum at S10 (*n* = 16). The average count of *E. faecium* (Enterococci) was 107. The corresponding OTU counts of the fecal and non-fecal coliforms, enterococci, and the ESKAPEE pathogens are presented in Table 3. The

OTUs of genera critically acclaimed by WHO such as *S. aureus* (mean = 85), *A. baumanii* (mean = 334), and *P. aeruginosa* (mean = 832) were also found across all the marine sediments of Kuwait. Intriguingly, detection of the highest counts of *P. aeruginosa* (n = 5590), *K. pneumonia* (n = 362), and *Enterobacter* sps. (n = 841) at S12 warrants further investigations at this location and adjoining areas. It would also be interesting to examine the ARGs hosted within these pathogens.

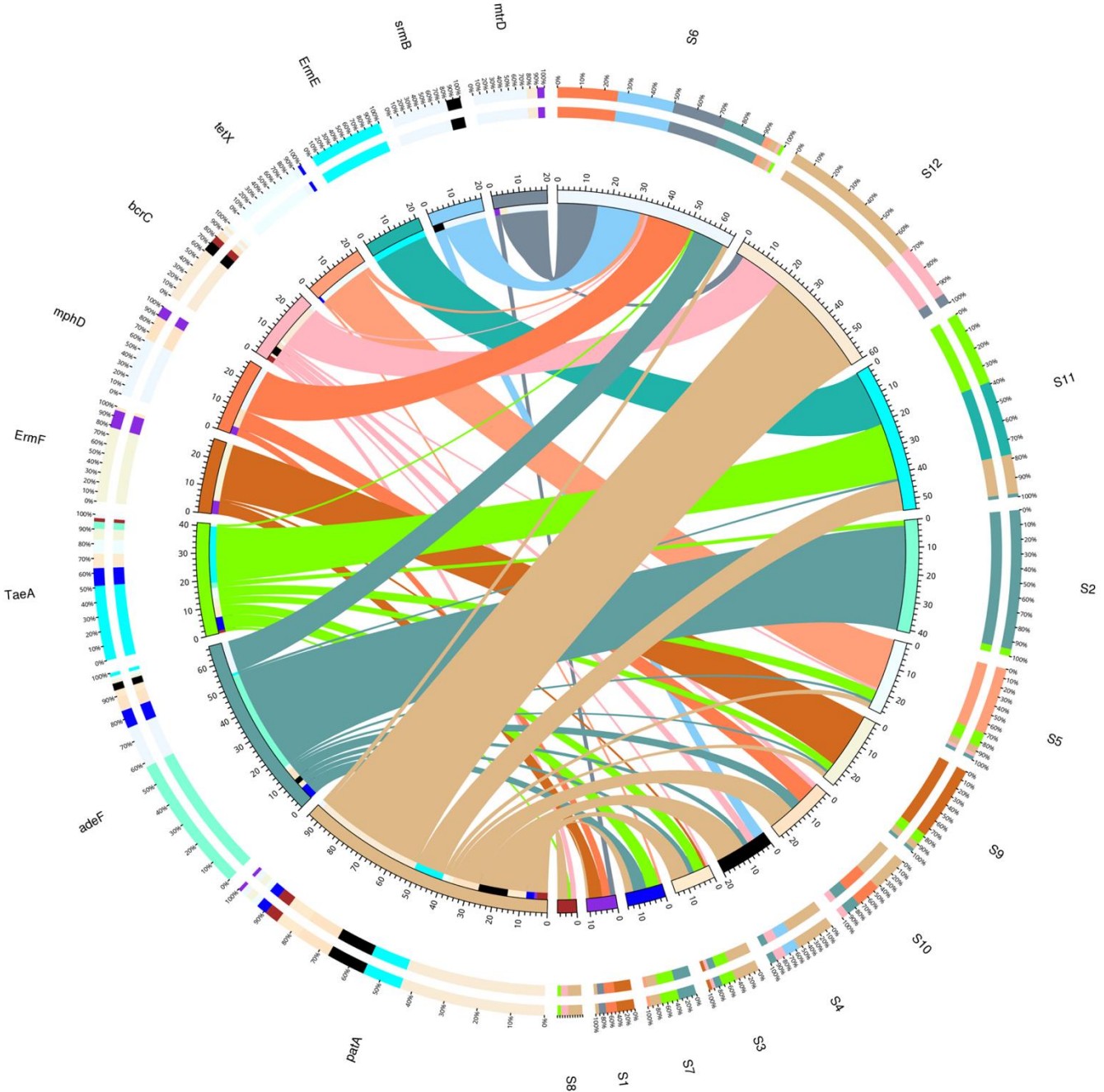

**Figure 3.** Circos plot showing the abundance of the dominant ARGs. The right side shows the sampling locations, and the left side shows the ARGs. Different colors in the inner circle represent different samples (RHS) and ARGs (LHS). Bar lengths at the LHS of the outer circle show the relative percentage of ARGs, and at the RHS show the relative percentage of the sample in which the antibiotic resistance gene was located.

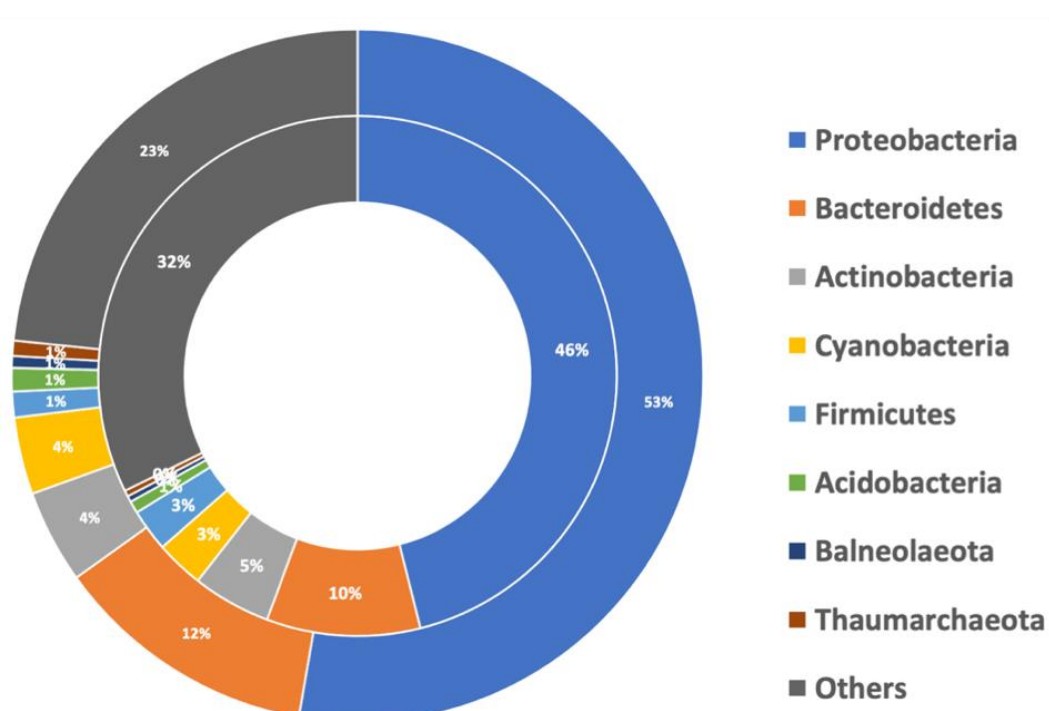

**Figure 4.** Double-doughnut chart representing the major phyla hosting the resistance genes in the marine sediments. The outer circle shows the distribution of phyla associated with all the bacterial genes, whereas the inner circle illustrates the dominant phyla bearing the ARGs.

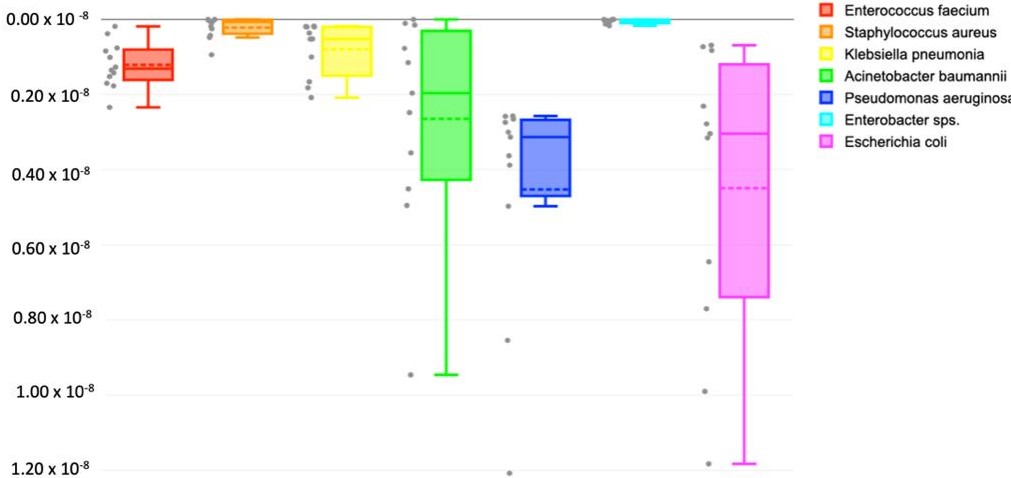

**Figure 5.** Box plots showing the relative abundances of ESKAPEE pathogens in the surface sediments collected from Kuwait's coast. Each box represents the interquartile range (25–75%), upper and lower whiskers are −10–90%, and dashed lines denote the mean RA%. The dots represent the RA at 12 sampling points. The *x*-axis shows the genera, and the corresponding RA is plotted on the *y*-axis.

### 3.4. ARGs against Drug Classes

We observed that the genes were active against 34 drug classes. The majority were resistant to the beta-lactams, cephalosporins (102), and penam (102). In addition, ARGs resistant to tetracycline (93), aminoglycoside (63), fluoroquinolone (55), carbapenem (48), lincosamide (40), cephamycin (35), phenicol (34), streptogramin (34), peptide antibiotics (29), and monobactum (28) were also recorded (Figure 6). A total of 25 or fewer ARGs were resistant to the drug classes of the glycopeptide, aminocoumarin, diaminopyrimidine, penem, rifamycin, glycylcycline, acridine dye, triclosan, pleuromutilin, sulfonamide, fos-

fomycin, nucleoside antibiotic, oxazolidinone, fusidic acid, mupirocin, sulfone, elfamycin, nitroimidazole, antibacterial free fatty acid antibiotics, and nitrofuran. There were eight ARGs not assigned to specific drug classes (unknown). Further investigations to identify these cryptic drug classes are highly recommended. Approximately 42% of the ARGs were resistant against two or more drug classes.

**Table 3.** Absolute OTU counts of ESKAPEE pathogens detected across Kuwait's coast.

| ESKAPEE Pathogens | Absolute Abundance (OTUs) | | | | | | | | | | | | Average |
|---|---|---|---|---|---|---|---|---|---|---|---|---|---|
| | **S1** | **S2** | **S3** | **S4** | **S5** | **S6** | **S7** | **S8** | **S9** | **S10** | **S11** | **S12** | |
| *Escherichia coli* | 3996 | 64 | 680 | 204 | 874 | 569 | 73 | 246 | 1044 | 268 | 278 | 61 | 696 |
| *Klebsiella pneumoniae* | 161 | 18 | 18 | 147 | 46 | 88 | 46 | 32 | 183 | 16 | 17 | 362 | 94 |
| *Staphylococcus aureus* | 0 | 22 | 5 | 16 | 1 | 42 | 83 | 807 | 5 | 0 | 37 | 1 | 85 |
| *Pseudomonas aeruginosa* | 265 | 227 | 242 | 1067 | 439 | 277 | 320 | 234 | 754 | 228 | 343 | 5590 | 832 |
| *Acinetobacter baumannii* | 835 | 13 | 398 | 102 | 314 | 437 | 9 | 68 | 0 | 1558 | 219 | 173 | 344 |
| *Enterococcus faecium* | 74 | 34 | 67 | 157 | 89 | 120 | 16 | 134 | 125 | 151 | 207 | 112 | 107 |
| *Enterobacter sps.* | 11 | 0 | 28 | 67 | 26 | 0 | 10 | 26 | 0 | 8 | 15 | 841 | 86 |
| Total | 5343 | 377 | 1438 | 1759 | 1789 | 1534 | 558 | 1547 | 2112 | 2229 | 1116 | 7138 | 2245 |

Non-fecal coliform: *Escherichia coli*; fecal coliforms: *Enterobacter* sps., *K. pneumonia*; enterococci: *E. faecium*. *S. aureus*, *P. aeruginosa*, and *A. baumanii* are on the WHO list of pathogens with multidrug resistance.

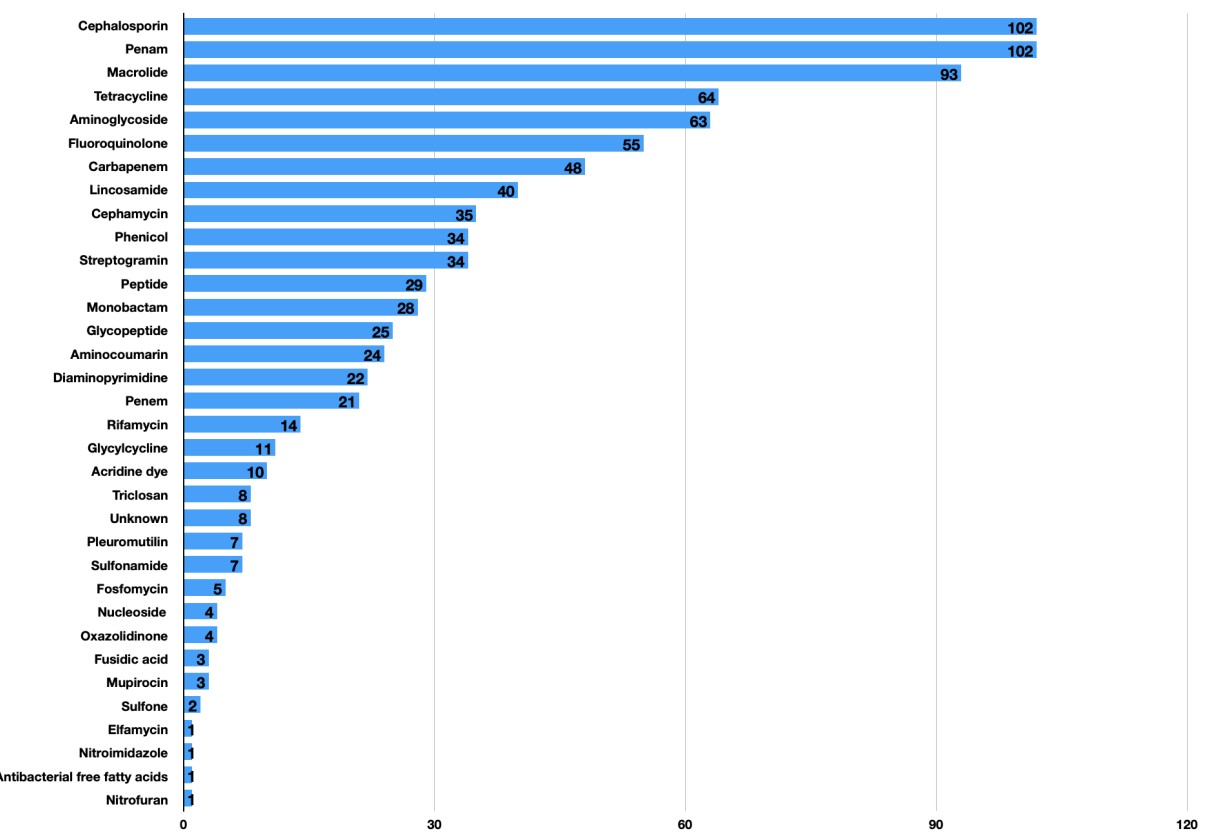

**Figure 6.** Bar graph showing the antibiotic drug classes of the identified ARGs. The *x*-axis presents the count of the genes, and the corresponding drug class is plotted on the *y*-axis.

### 3.5. Mode of Action of ARGs

Further assessment of the mode of actions corresponding to the predicted ARGs revealed a majority of them were acting by antibiotic efflux, followed by antibiotic inactivation, antibiotic target alteration, antibiotic target protection, antibiotic target replacement, and reduced permeability toward the antibiotics (Figure 7). A few of them also acted through multiple modes such as antibiotic efflux and reduced permeability to the antibiotic, antibiotic target alteration and antibiotic inactivation, and antibiotic target alteration and antibiotic target replacement.

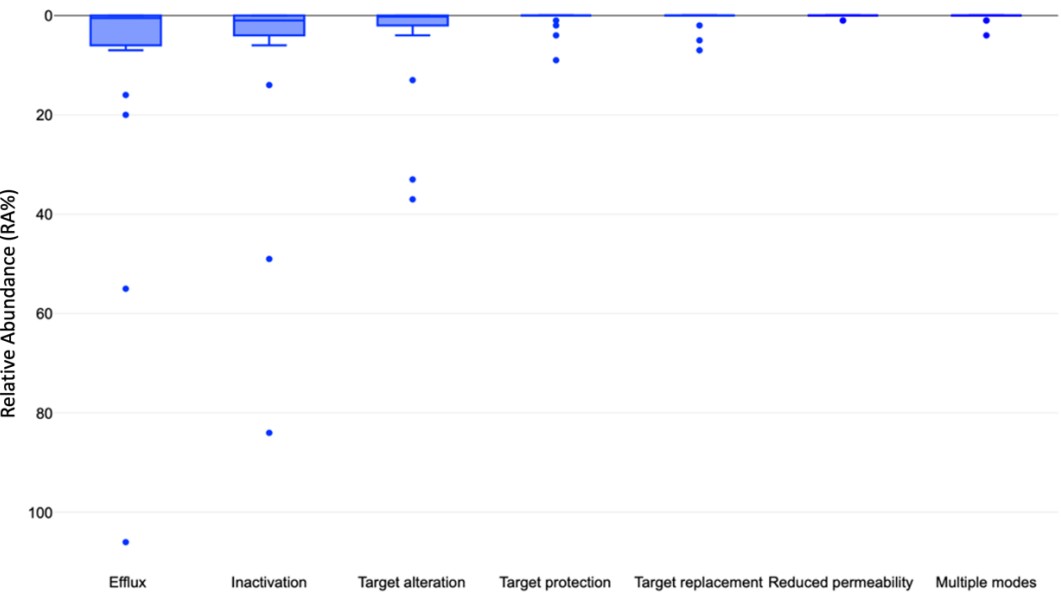

**Figure 7.** Modes of action of ARGs discovered in marine sediments from Kuwaiti shores. Each box represents the interquartile range (25–75%), upper and lower whiskers are −10–90%, and dashed blue lines mean RA%.

### 3.6. Intersite Variability of ARGs

We compared the distribution of ARGs between Group A (S1, S2, S3, S5, S6, S7, S8, S9, S10, and S11) and Group B (S4 and S12). Group A comprised locations that were near the outfalls, whereas Group B included the pristine and clean sampling areas. An average of 120 ARGs were observed in Group A, which was almost 1.7 times higher than in Group B (mean: 70) (Figure 8a). Analysis of unique and common ARGs revealed that approximately 80 genes were common between both groups. Group A possessed 290 unique ARGs, while 32 were included in Group B (Figure 8b). We believe the higher numbers of genes in Group A were due to higher exposure of these microbes to dissolved antibiotics discharged through the outfalls than in Group B, which were cleaner sites. The hierarchical clustering also revealed that the prevalence of ARGs differed spatially (Figure 8c). The hierarchical clustering also revealed that ARGs were spatially different (Figure 8c). The prevalence of ARGs varied at each site within the same group. Metastats revealed vanHF to be significantly different between Group A and Group B (q < 0.05).

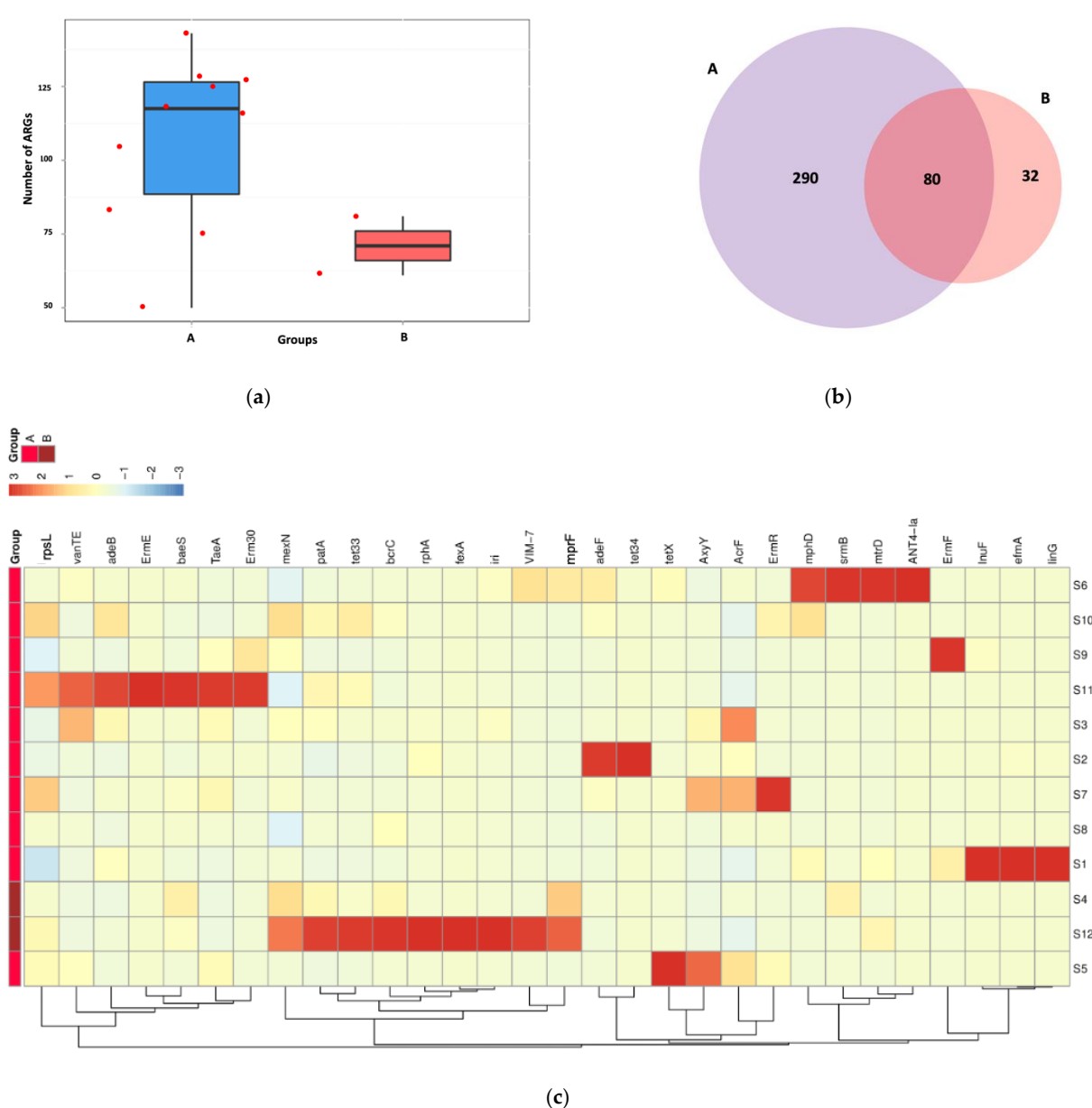

**Figure 8.** (**a**) Number of ARGs distributed between Group A and B; (**b**) Venn flower diagram showing common and unique ARGs between group A and B; (**c**) hierarchical clustering of ARGs.

## 4. Discussion

The detection of pharmaceuticals, including antibiotics, in marine water draws attention to the risk of ARGs in marine biota living in close proximity to these outfalls. This study highlighted the abundance of ARGs within different phyla found in the marine sediments through shotgun metagenomic sequencing.

In the present samples, a total of 402 ARGs were identified, which was significantly less than the 819 ARGs found in sediments near coral reefs of Xisha Island [49], and the 2354 ARGs from the Gulf of Khambat, India [50]. On the other hand, a study of the Lonar soda lake in India reported a mere 26 ARGs [51] in the surface sediments. The variation in ARG abundance can be related to the type and scale of anthropogenic pollution, as it can be a factor enhancing and disseminating ARGs in the surface sediments [52]. Chemical pollutants and the different fractions of metals in the sediments are also significant drivers of ARG accumulation in the sediments [25,53]. Some likely sources of ARGs in Kuwait's marine environment could be the chronic metal discharges from the shipping industry,

desalination plants, and airborne input with dust, in addition to the pharmaceuticals introduced through wastewater discharges. The AMR is also reported in the Yellow River Delta [54], Yangtze river basin [55], Karst River [56], Xisha Island [49], and Ili River [57]. Proteobacteria (46%), Bacteroidetes (10%), and Actinobacteria (5%) were the main hosts of ARGs in the Kuwait coastal area. The high abundance of Proteobacteria is likely because they are the most flexible metabolically and are capable of adapting to fluctuating environments.

In agreement with previous studies, *E. coli* derived from Kuwait seawater and biota sampled across seasons displayed high antibiotic resistance against ampicillin (70% and 69%, respectively) [4]. These samples were collected from three sites in Kuwait and Bahrain and four sites from Oman and the United Arab Emirates between December 2018 and May 2019; ampicillin resistance rates were 29.6% for *E. coli* isolated from seawater samples [58]. Light et al. [58] reported whole-genome sequencing on a subset of 173 *E. coli* isolates, and high carriage rates of qnrS1 (60/173) and blaCTX-M-15 (45/173) were observed, correlating with reduced susceptibility to the fluoroquinolones and third-generation cephalosporins. *E. coli* was one of the genera detected among the other ESKAPEE pathogens in the present study. In addition to this, lower abundances of coliforms and enterococci in the coastal sediments of Kuwait cannot be ignored. Their contribution toward AMR needs to be further investigated. The methicillin-resistant *S. aureus,* vancomycin-resistant *E. faecium,* carbapenem-resistant *A. baumanii,* and *P. aeruginosa* are on the WHO global priority pathogens list of antibiotic-resistant bacteria [59].

In the present samples, patA was the most dominant gene, followed by adeF, rpsL, TaeA, ermF, and tetX. Our results were in partial agreement with a study conducted in the deep sediments of the Mariana Trench, in which ermF, tetM, tetQ, cfxA2, PBP-2X, and PBP-1A were common [52]. The most abundant gene in the Ili river was adeF [57]. The distribution of ARGs spatially was variably attributed to the differences in antibiotic usage in those regions. The genes discovered in the present study exhibited resistance against fluoroquinolone, tetracycline, streptogramin, macrolide, lincosamide, pleuromutilin, glycylcycline, a peptide antibiotic, a phenicol antibiotic, an oxazolidinone antibiotic, penam, aminocoumarin, aminoglycoside, cephalosporin, carbapenem, cephamycin, and glycopeptide antibiotics. Many of these antibiotics were previously reported in Kuwait's coastal areas [22], as well in as the wastewater streams of Al Kabd and Umm ul Hayman in Kuwait [23].

The predominant ARGs belonged to 13 AMR gene families; this could be related to the prolonged antibiotic exposure resulting in an antibiotic resistome due to continuous coevolution of small molecules in the environment and microbial genomes. This is almost comparable to 17 gene families discovered at Tolo Harbour in the South China Sea [30]. The RA of all the genes varied according to the sampling location [50,60,61]. In the present study, the presence of ARGs was also recorded in sediments from sites S4 and S12 that were not in proximity to outfalls. This could be supported by the view that bacteria naturally produce antibiotics as a mechanism of defense [29] and that these ARGs can be metal-mediated, which is dominant across the marine environment. The primary mechanism of action in Kuwait was antibiotic efflux. The antibiotic resistome can also have cryptic resistance genes that might occur due to chronic exposure to antibiotics, metals, and biocides [62]. This is a highly preferred way of acquiring bacterial resistance, as the efflux pumps recognize a variety of substrates that are expressed in a broad range of pathogens [63].

## 5. Conclusions

This baseline study underlined the occurrence of a diverse pool of ARGs in the coastal sediments of Kuwait. How these ARGs influence the health of the marine ecosystem needs to be systematically assessed on a larger spatial scale. Continuous monitoring and sampling at additional sites at different depths and seasons need to be carried out to ensure sustainability. To track the dissemination of ARGs in the marine ecosystem and

their influence on human health, the mobile genetic elements and integrons should be investigated in the future.

**Supplementary Materials:** The following supporting information can be downloaded at: https://www.mdpi.com/article/10.3390/su14138029/s1, Table S1: ARGs found in marine sediments of Kuwait, Supplementary S1. Figures S1–S12: Library QC of 12 marine sediment samples collected from Kuwait Supplementary S2.

**Author Contributions:** Conceptualization, N.H. and S.U.; methodology, A.S., N.A.R. and F.Z.; software, N.H.; validation, B.L., H.A.A.-S. and M.B.; formal analysis, S.U. and M.B.; investigation, H.A.A.-S. and B.L.; resources, F.A. and M.B.; writing—original draft preparation, N.H. and S.U.; writing—review and editing, N.H., S.U. and B.L.; visualization, N.H. All authors have read and agreed to the published version of the manuscript.

**Funding:** The APC was funded by the Kuwait Institute for Scientific Research.

**Institutional Review Board Statement:** Not applicable.

**Informed Consent Statement:** Not applicable.

**Data Availability Statement:** The raw sequences of this data are deposited in the public repository of the National Centre for Biotechnology Information under the accession number PRJNA819259 (SRR18461109-SRR1846120). The data can be accessed through the following link: https://dataview.ncbi.nlm.nih.gov/object/PRJNA819259?reviewer=mbemfgsv8tgn5as9f8m8hrorbe (accessed on 12 June 2022).

**Acknowledgments:** The authors are thankful to the Kuwait Institute for Scientific Research for supporting this study.

**Conflicts of Interest:** The authors declare no conflict of interest.

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
