# Peer review of "Antibiotic Resistance Genes Associated with Marine Surface Sediments: A Baseline from the Shores of Kuwait"

_sustainability, doi:10.3390/su14138029_

Round 1

Reviewer 1 Report

This paper comprehensively examines antibiotic resistance genes (ARGs) of DNA collected form from marine sediments by metagenome analysis. They obtain information on the distribution of ARGs in marine sediments. Methods and analysis are standard, and it is judged that the obtained data can be trusted. However, it is necessary to present the amount of DNA recovered from the sediment and clarify the detailed conditions for metagenomic analysis. These two points are indispensable for obtaining the credibility of this paper. The discussion was based on the results obtained and no arbitrary content was confirmed. Bacterial count and water quality information in the surrounding the sampling sites are missing. At the least, it is desirable to obtain information on the number of E. coli, coliforms, and enterococci in the marine sediments.

Due to the lack of information on the resistance genes of marine sediments at present, this paper can be a valuable source of information.

Some important points that need to be considered are listed as follows:

1) 2.1. Sample Collection and DNA Extraction

It is necessary to indicate the recovered amount of DNA and the concentration of the DNA sample used for the sequencing.

2) 2.2. Metagenomic Sequencing

The specific method and analysis method examined for metagenomic sequencing in this study should be explained concretely. It is too general to confirm the reliability of the analysis results.

3) 3.2. Antibiotic resistance gene profiles

throughout the whole, the abbreviations for ARGs are not written correctly. For all of the text, Figs, and Tables, it is necessary to unify abbreviations for each ARG and match the description in the text, Figs, and Tables.

4) The resolution of Fig. 3 is low. Since it is an important figure, display it clearly.

5) Line 248­–255

The ARGs have been detected in the S4 and S12, which were set as sites where there was no direct source of pollution. In particular, various ARGs have been detected in high concentrations in the S12. It is considered to be the most important point in the survey results of this study. I would like you to consider not the general simple consideration based on different sites in China, but the local characteristics in the Shores of Kuwait.

Reviewer 2 Report

This manuscript refers on the occurrence and distribution of antibiotic resistance genes in coastal marine sediments of the Kuwait region. This subject is of high interest for the scientific community and several literature reports have currently been published on this theme. Although not fully original, in the examined area general data on the presence of contaminants have been reported (as correctly cited by the authors), therefore I think that this further study could be of added value. Some minor changes are suggested to clarify the text, as reported below.

Title: Antibiotic Resistance Genes associated to marine....

(not resistant, please also in the text check this). English language revision by a native English speaker is also suggested. 

Abstract, lines 15- 16, The presence

line 17, was recorded........ the prevalent were... (not most prevalent)

line 19, showing resistance against  (not offering)

line 20, maximum resistance was detected

About 46% of gene....this sentence could be moved close that of 46% of genes originated from the phylum Proteobacteria.

line 24, All the ARGs (S not in capitals)

line 26. coastal marine sediments (not ocean environment)

line 27, dissemination to surrounding ecosystems

line 44, are reservoirs of ARGs (not deposited) that can be disseminated in marine environments  

Before the aims of this study a sentence regarding the occurrence of antibacterial resistant bacteria should be introduced (see the review by Marti...Balcazar, 2014. The role of aquatic ecosystems as reservoirs of antibiotic resistance)

Fig. 1 the resolution of this figure is low, should the authors improve it?

At stations S10 and S4 two dot points should be visible

In the materials and methods the softwares used for the creation of the Double doughnut chart, Venn diagramas and hierarchical clustering  must be indicated (like what the authors made for circos plots. 

lines 89 and 92, the exponential numbers e-5 should have been reduced in size

Results. I suggest to move lines 97-103 in the materials and methods section

line 100 Klbebsiella pneumoniae

line 101, Acinetobacter and Enterobacter spp. ; the name of these bacteria must be italicized

line 121, These genes belonged to the phyla.....please indicate

Table 2 Lenght (in full)

Caption to Figures 2 and 7, Relative abundance (RA%)

line 145, The major phyla

line 147, % of the total

line 152, to evaluate (instead of to look at)

Caption to fig. 4, distribution of phyla hosting  the resistance genes

line 161. denote; in fig. 4 the numbers are hardly visible, increase their size

3.4 ARG against Drug classes

line 165, The majority were resistant (instead of Maximum were resistant)

line 174, ARGs not assigned to specific drug classes

line 193, Group A comprised locations that...

line 197, common between both groups.... while 32 were included in Group B

line 200, revealed that ARG were spatially different

line 209, within different phyla

line 218, Some likely sources .. could be the chronic metal discharges

line 222. In agreement with previous studies, E. coli isolated

line 223, displayed high antibiotic resistance against ampicillin

Please explain if both studies (4) and (46) refer to the same samples why there is so high variability in seawater (70% versus 29.6%)

Reference to the paper by Light et al. (2022) must be included in the reference list

line 251, the concept of a natural resistome should be included (Wright, G. The antibiotic resistome: the nexus of chemical and genetic diversity. Nat Rev Microbiol 5, 175–186 (2007). https://doi.org/10.1038/nrmicro1614)

line 252, the primary mechanism of action... was 

line 257, baseline study underlines the occurrence

line 259, continuous monitoring

The take-home messages for SUSTAINABILITY should be underlined better.

Author Response

Response to the reviewers comments are provided in the attached file
